# Multi-Device Piezoelectric Direct Discharge for Large Area Plasma Treatment

**Dariusz Korzec \*** , **Florian Hoppenthaler, Anatoly Shestakov, Dominik Burger, Andrej Shapiro, Thomas Andres, Simona Lerach and Stefan Nettesheim**

Relyon Plasma GmbH, Osterhofener Straße 6, 93055 Regensburg, Germany;
f.hoppenthaler@relyon-plasma.com (F.H.); a.shestakov@relyon-plasma.com (A.S.);
d.burger@relyon-plasma.com (D.B.); a.shapiro@relyon-plasma.com (A.S.);
t.andres@relyon-plasma.com (T.A.); s.lerach@relyon-plasma.com (S.L.);
s.nettesheim@relyon-plasma.com (S.N.)
**\*** Correspondence: d.korzec@relyon-plasma.com

**Abstract:** The piezoelectric cold plasma generators (PCPG) allow for production of the piezoelectric direct discharge (PDD), which is a kind of cold atmospheric pressure plasma (APP). The subjects of this study are different arrays of PCPGs for large-area treatment of planar substrates. Two limiting factors are crucial for design of such arrays: (i) the parasitic coupling between PCPGs resulting in minimum allowed distance between devices, and (ii) the homogeneity of large area treatment, requiring an overlap of the activation zones resulting from each PCPG. The first limitation is investigated by the use of electric measurements. The minimum distance for operation of 4 cm between two PCPGs is determined by measurement of the energy coupling from an active PCPG to a passive one. The capacitive probe is used to evaluate the interference between signals generated by two neighboring PCPGs. The second limitation is examined by activation image recording (AIR). Two application examples illustrate the compromising these two limiting factors: the treatment of large area planar substrates by PCPG array, and the pretreatment of silicon wafers with an array of PCPG driven dielectric barrier discharges (DBD).

**Keywords:** atmospheric plasma; resonant piezoelectric transformer; piezoelectric direct discharge; plasma arrays; surface treatment

## 1. Introduction

The upscaling of plasma systems has long been a concern of researchers [1–3]. Cold atmospheric pressure plasma jets (APPJ) are very important tools for surface processing [4–7]. Typically, the APPJs produce plasma small in size, from a few millimeters to a few centimeters. To up-scale the size of the treated substrate, different discharge architectures are applied based on the multiplication of a single plasma source. One example of such a methodology is the matrix of micro-hollow cathode discharges (MHCD) [8,9]. The arrays of atmospheric pressure plasma jets (APPJs) are subject of a number of papers [6,10–13]. Such array sources powered with frequency in the kHz range, based on dielectric barrier discharge (DBD) are described in [14,15].

Recently, the operation of piezoelectric direct discharge (PDD) [16,17] as an APPJ was characterized [18,19]. The resonant piezoelectric transformers (RPT) [20] used for generation of the PDD, can be used in arrays for the increase of the treatment area and speed. One approach is to build a multichannel RPT, as described in [21,22]. The disadvantage of such a solution is the technological complexity and the limitation to a specific task. More flexibility allows an approach based on modularity: arranging many single PDD generators in an array. Such an approach was demonstrated for six separate RPTs for a high power plasma generator [23]. The system for an arbitrary number of parallel-connected piezoelectric cold plasma generators (PCPGs) [24] is described in [25]. The problem faced

by practical realizations is the electrodynamic and acoustic interference of the PCPGs positioned close to each other. To avoid such interferences, a minimum distance between the PCPGs must be kept. On the other hand, for the homogeneous treatment of a broad moving substrate PCPG activation traces are needed with distances, which are smaller, than the activation width of a single PCPG. This width can be influenced either by the operating parameter such as PCPG power, distance between the PCPG and the substrate, the substrate movement speed and gas mixture, or by definition of the activation, e.g., reaching of some threshold value of the surface free energy. In this work, the activation image recording (AIR) system is used for the evaluation of the activation area.

The minimum distance for different types of PCPG is investigated by measurement of energy coupling from an active PCPG to a passive PCPG placed at some distance from the active one. The AIR experiments are used to determine the minimum required distance between PCPG traces for different device configurations. The resulting system architectures and following application examples are described and discussed.

## 2. Experimental Details

For all experiments presented in this study, two types of PCPGs were used: the CeraPlas™ F and CeraPlas™ HF. Their physical operation principle and the method of parameter control is described in [26]. Two discharge configurations were used. The first was the PDD. The second was the PDD-powered dielectric barrier discharge (DBD) [17].

### 2.1. Energy Transfer Measurement

The electric field generated by one PCPG can induce the electromechanical oscillations of another PCPG positioned in the vicinity of the first one, providing the resonance frequencies of both devices are very similar. This effect was used in this study for determining the minimum safe distance between two PCPGs working in an array of PDD devices. For this purpose, two PCPGs of CeraPlas™ HF type were positioned with their high voltage sides face-to-face. The first of them was not movable. The second one can be moved linearly by the micrometric manipulator to set the exact distance between the tips of the PCPGs.

Figure 1a shows the setup used for evaluation of the energy coupling between two PCPGs. A sinusoidal signal generator was used to generate the input signal of the first PCPG. The input terminals of the second PCPG were not powered but bridged by a $R_{out} = 10$ kΩ resistor. They were used as output terminals of the measurement setup. On input and output, the current and voltage were measured. For current measurements, the current probe Tektronix TCP202 was used. The voltage was measured by the use of the Tektronix P6015A voltage probe. Both probes were connected to the Tektronix DPO3034 phosphor oscilloscope (5 M record length; 2.5 GS/s sample rate).

### 2.2. Capacitive Probe Measurement

The high voltage tip of the PCPG was made of PZT ceramics and was not equipped with any electrically conducting electrode. Consequently, the direct voltage measurement of the PCPG output voltage was impossible. However, during operation, the PCPG produces a strong alternating electric field causing the plasma ignition. This field can be evaluated by the use of the large area capacitive probe. The details about this method and the signal interpretation were described in detail in [19]. The voltages measured by such probe are proportional to the kHz voltage on the tip of the PCPG. The proportionality factor is in the $10^{-3}$ range. In this study, the capacitive large area probe was used to investigate the operation of a couple of PCPG working at a close distance from each other, as shown in Figure 1b.

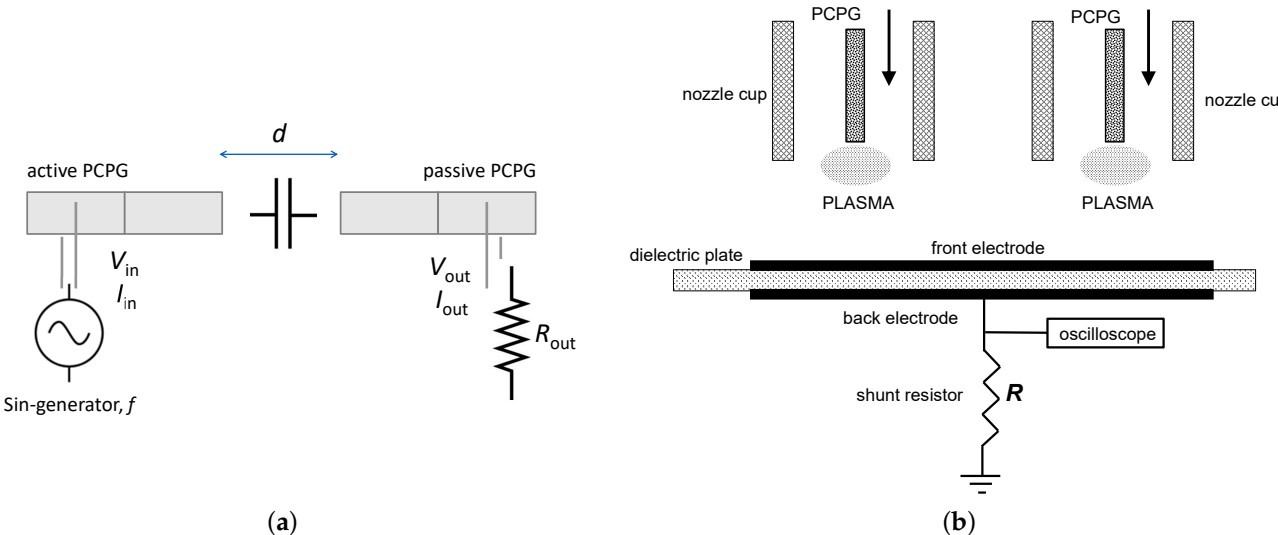

**Figure 1.** The setup for electric characterization by use of: (**a**) energy transfer efficiency between two PCPGs, and (**b**) capacitive probe measurement.

### 2.3. Activation Area Determination

To determine the activation area of a PCPG, the device was positioned to assure a required distance between the PCPG tip and the substrate surface, typically 4.5 mm. The substrate used for evaluation is the HDPE plate 2 mm × 50 mm × 100 mm. The usual treatment time is 10 s for open PDD and 20 s for the PDD-powered DBD.

The test ink was used for visualization of the activated zone. To avoid the aging effect due to stechiometry changes of multi-liquid test ink, the 58 mN/m test ink (Fa. Ahlbrandt) consisting of pure formamide is used. The additional advantage of the value 58 mN/m was that it assures a high dynamic range of the evaluation method, because it was positioned in the middle between the surface energy value of non-treated HDPE (36 mN/m) and the maximum reachable value for a plasma-treated surface, of 72 mN/m.

It can be observed that, after covering the activated surface with test ink, the ink spot area decreased within seconds. The pictures of the ink spots were taken in short intervals using a digital camera. The contour of the ink spot was automatically recognized and the ink spot area was calculated from the pixel count using specialized software. The system used for this purpose was the Activation Image Recording (AIR) of Relyon Plasma GmbH. From the collected shrinkage curve, showing the activation area as a function of time, the point after 10 s of shrinking was selected as a reference value. The AIR method is described in detail in [19].

### 3. Results and Discussion

#### 3.1. Interferences between PCPGs

Three types of interferences between two PCPGs working in small distance from each other can be observed. The first one occured, if the PCPGs operated at high power are very close to each other, with a tip-to-tip distance of less than 30 mm. In such a case, the spark discharge between the PCPGs can be observed which disrupts the regular operation of both PCPGs. Working for a long time in such a mode can cause damage to the devices.

The second one is based on the acoustic energy transfer between the PCPGs [27]. Since the condition of acoustic impedance matching are not fulfilled in our setup, only a small part of the energy coupled in the PCPG can be transferred acoustically to another PCPG. Nevertheless, the acoustic noise, depending on the control drive used, can be observed for the tip-to-tip distance up to 8 cm. The most probable reason for the acoustic noise is the beat between the ultrasonic frequencies of both PCPGs. The operation frequencies of the PCPGs are very similar, but not identical. The reasons for this difference can be related

either to the device characteristics or to the operating conditions. The most important device-related influence has the length of the PCPG and its electric parameters. They have a production process-related statistical distribution, which causes the differences in the resonant frequencies. The operation related difference in frequency is caused by the PCPG control system. To keep the power of the PCPG constant, the frequency was varied. This control mechanism was described in detail in [26]. The overlapping of two signals with operation frequencies $f_1$ and $f_2$ generated by two neighboring PCPGs respectively, results in a beat frequency $f_b$ [28] given as:

$$f_b = |f_2 - f_1|. \tag{1}$$

The $f_b$ is typically in the small kHz range and is audible. It is especially loud, if some acoustic resonances in the holder and connections of the PCPG are excited. The source of acoustic noise can also be the excitation of transversal resonances in the PCPG itself. Since the transversal sound velocity in hard PZT is a factor two lower than the longitudinal one, and the first harmonic frequency can be excited, the transversal resonance frequency is a quarter of the second harmonic frequency of the PCPG. For CeraPlas™ F it is 50 kHz/4 = 12.5 kHz. This was a high pitch sound, which was frequently audible if two PCPGs are operated close to each other. Such noise is strong enough to exclude many attractive applications of the PCPG arrays.

The third type of interference was by the electric fields of the both PCPGs and will be investigated in more detail in the following sections. It is effective even if the PCPG oscillation amplitude is too weak to cause the spark discharge or the loud noise.

### 3.2. Coupling Efficiency

The active PCPG, as described in Section 2.1, is operated with the input voltage of 1 V. The voltage transformation ratio for PCPG was about 1000, resulting in an output voltage of the active PCPG in the range of 1000 V, which is not sufficient to cause a gaseous breakdown at the PCPG tip. Consequently, the electric coupling between the active and passive PCPG is purely capacitive, with no losses for the discharge. Since no parasitic discharge between PCPGs can ignite, the measurement can be conducted starting from very small distances between the PCPGs. Figure 2a shows the voltage induced in the passive PCPG as a function of this distance. The output voltage for distance below 3 mm is larger than the corresponding input voltage. The reason can be the higher oscillation quality of the second PCPG, compared with the first one. The difference in oscillation resonance quality can be caused by the different input impedance of the first PCPG and the load of the second one.

The output voltage decreases rapidly with increasing distance and vanishes for the distance larger than 3 cm. The coupling efficiency, defined as the output-to-input power ratio, is decaying for the distance of more than 3 cm, as shown in Figure 2b. However, it reaches the maximum value of 80.4% at the distance of 2.4 mm. Further increase of the coupling efficiency can be achieved by precise adjustment of the load resistor. With decreasing coupling efficiency, the resonance frequency of the active PCPG is increasing, reaching its non-loaded frequency by a distance of about 15 mm (see Figure 2b). The practical conclusion from this investigation is that the CeraPlas™ HF devices placed in distances larger than 3 cm will not cause the electric interference between them.

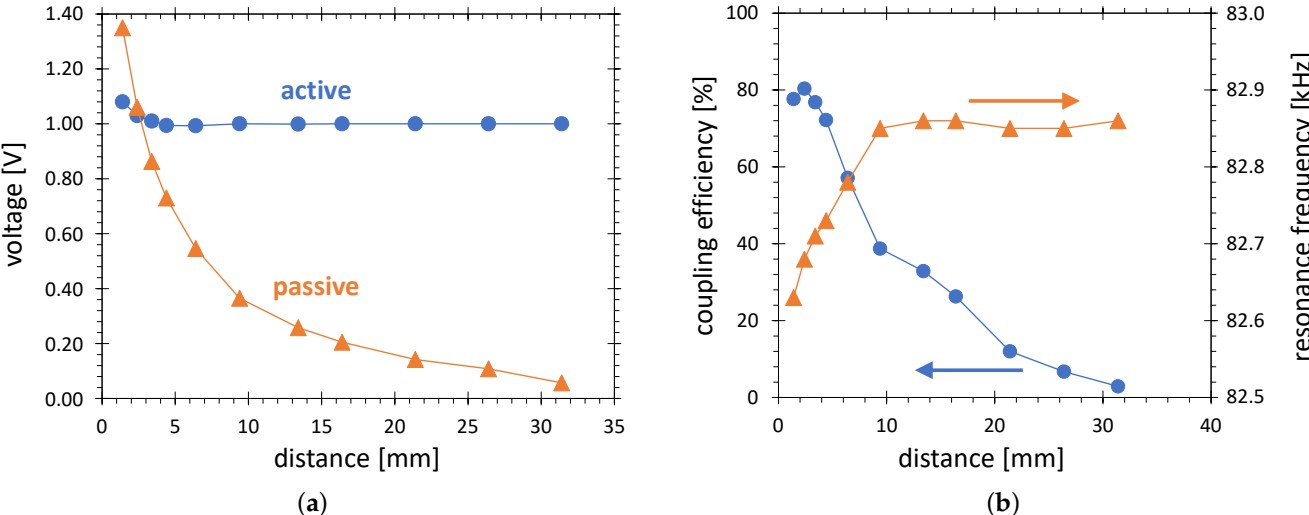

**Figure 2.** The energy coupling from active to passive PCPG (see Figure 1a) as a function of distance between the PCPG tips. (**a**) Voltages on the low voltage sides of the PCPGs. (**b**) The coupling efficiency and the resonance frequency of the active PCPG determined with 10 kΩ load on the low voltage side of the passive PCPG. Both PCPGs used are of CeraPlas™ HF type. The signal on input of the active PCPG was sinusoidal with amplitude of 1 V.

### 3.3. Overlapping of Emitted Signals

A typical signal measured by a capacitive probe for PCPG of CeraPlas™ F-type operated with 8 W power in the tip-to-probe distance of 30 mm is shown in Figure 3a. It can be interpreted as an overlap of a more-or-less sinusoidal signal and short nonperiodic pulses being a response on micro-discharges. The frequency of the sinusoidal component corresponds to the second harmonic of the PCPG oscillation. The positive and negative micro-discharge responses are different. Typically, the positive (anodic) peaks have a higher magnitude than the negative (cathodic) ones. Such asymmetry is due to the difference in the physics of the positive and negative streamers.

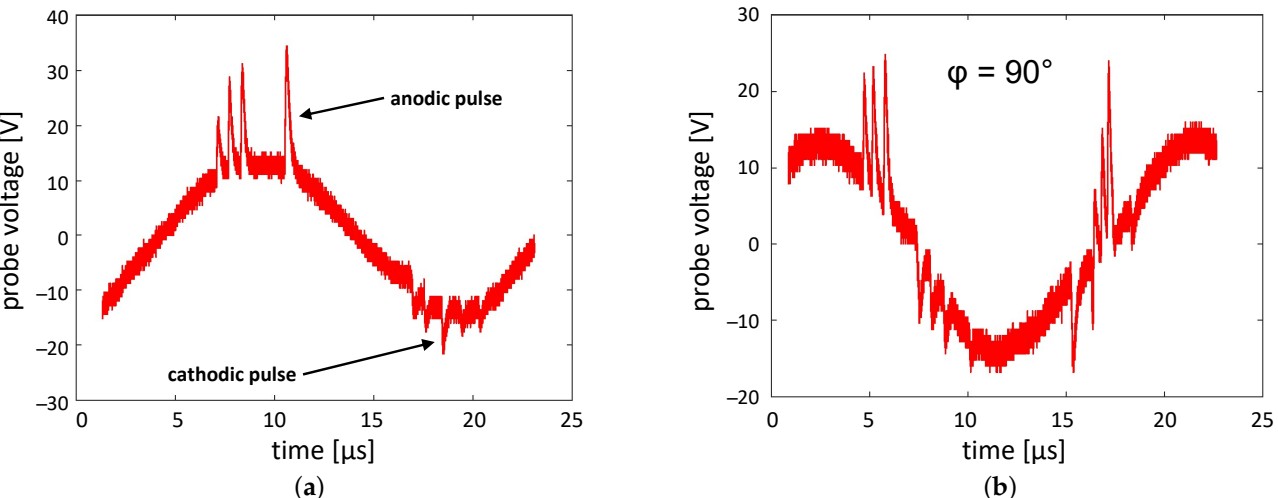

**Figure 3.** The probe voltage measured for: (**a**) single PCPG, and (**b**) two PCPGs placed parallel at a tip-to-tip distance of 60 mm, operated with the input power of 8.0 W and the CDA flow of 8 SLM each. The distance between the PCPG tip and the probe surface is 30 mm.

Figure 3b shows the probe voltage measured when two PCPG placed in the tip-to-tip distance of 60 mm are operated in front of the probe. At this distance no field interference between the PCPGs can be observed. The two signals are not coherent in phase. In curve in Figure 3b during the time corresponding to one cycle of the PCPG second-harmonic oscillation, two cathodic and two anodic micro-discharge peaks can be observed. The phase shift between the two independently oscillating PCPGs results in a shift of the micro-discharge peaks in respect to the maximum of the sinusoidal curve.

Due to slightly different oscillation frequencies of the both PCPGs, the phase difference of the two signals is not constant in time. Consequently, in another voltage excerpts the signals with 0° or 180° phase difference can be found. Examples of voltage curves for such phase shifts are shown in Figure 4a,b respectively.

When comparing the curves for a single PCPG in Figure 3a and for two PCPGs working in phase (phase difference of 0°) as shown in Figure 4a, it can be seen that the signal components related to the PCPG oscillations are added. The amplitude for the single PCPG is about 15 V. For two PCPGs working in phase, it is doubled to about 30 V. The cathodic and anodic micro-discharge peaks occur at about the same place as a single PCPG signal, but the number of micro-discharges doubles—in the shown examples eight instead of four anodic micro-discharges and 10 instead of five cathodic micro-discharges.

When comparing the curves for a single PCPG in Figure 3a and for two PCPGs working with a phase shift of 180° as shown in Figure 4b it can be seen, that the signal components related to the PCPG oscillation are compensating each other on the capacitive probe. The occurrence of the anodic micro-discharges of one PCPG is overlapping with the cathodic signals of another PCPG.

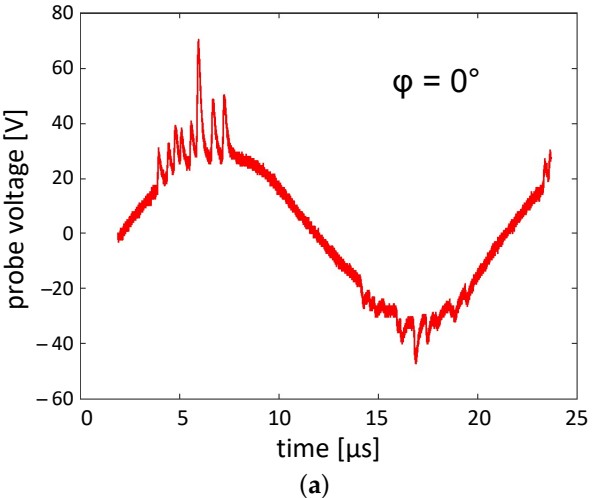

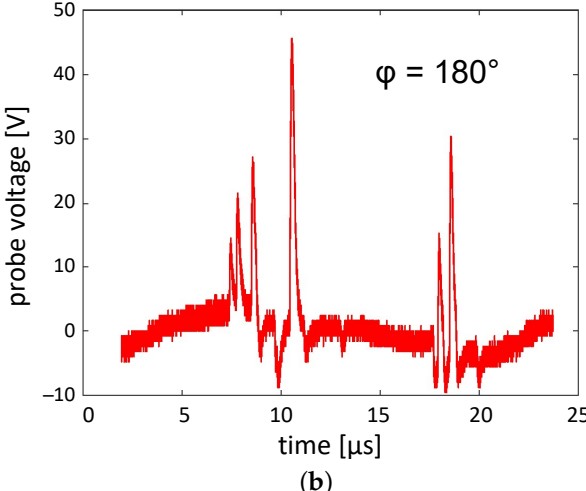

(**a**)  (**b**)

**Figure 4.** The probe voltage measured by capacitive probe for two concurrently working PCPGs with phase shift of (**a**) 0° and (**b**) 180°. The input power and CDA flow of the PCPGs is 8.0 W and 8 SLM respectively. The distance between PCPGs is 60 mm.

Such independence of the voltages for a couple of PCPGs of CeraPlas™ F type can be observed for tip-to-tip distances down to 40 mm. It can be concluded that at such sufficient distances, the PCPGs run independently and do not influence each other.

### 3.4. Surface Activation

The most important parameter of a plasma system used for surface activation is the activation rate, defined as the activated area per time unit. In case of static treatment, the activation rate $\eta_{\text{act}}$ can be expressed as the ratio of the activated area $S_{\text{act}}$ to the activation time $t_{\text{treat}}$:

$$\eta_{\text{act}} = \frac{S_{\text{act}}}{t_{\text{treat}}} \tag{2}$$

The activation image of a single PCPG operated statically with the power of 8 W generated on the HDPE substrate is shown in Figure 5a. The kidney shape of this image results from the structure of the PDD shown in the picture included in the upper-right corner of the figure. The Equation (2) yields for $t_{\text{treat}} = 10$ s and $S_{\text{act}} = 400$ mm² the activation rate of 40 mm²/s.

In case of the treatment with the relative speed $v_{\text{treat}}$ between the substrate and the PCPG, the Equation (2) can be rewritten as:

$$\eta_{\text{act}} = v_{\text{treat}} w_{\text{act}} \tag{3}$$

where $w_{\text{act}}$ is the width of the activation strip.

The rotational asymmetry causes the activation resulting in moving substrates depends on the movement direction. Figure 5b illustrates this dependence. Two curves represent the interaction length of a point on the substrate with the activation zone of PCPG moving relatively to the substrate surface. The position of the treated substrate point is defined on the axis perpendicular to the motion direction. The line with red bullets shows the result for substrate movement along the *x*-axis accordingly to the axis orientation shown in Figure 5a. The full width at half maximum (FWHM) of about 22.5 mm by such movement direction is by 40% larger than the FWHM for movement along the *y*-axis of 16 mm.

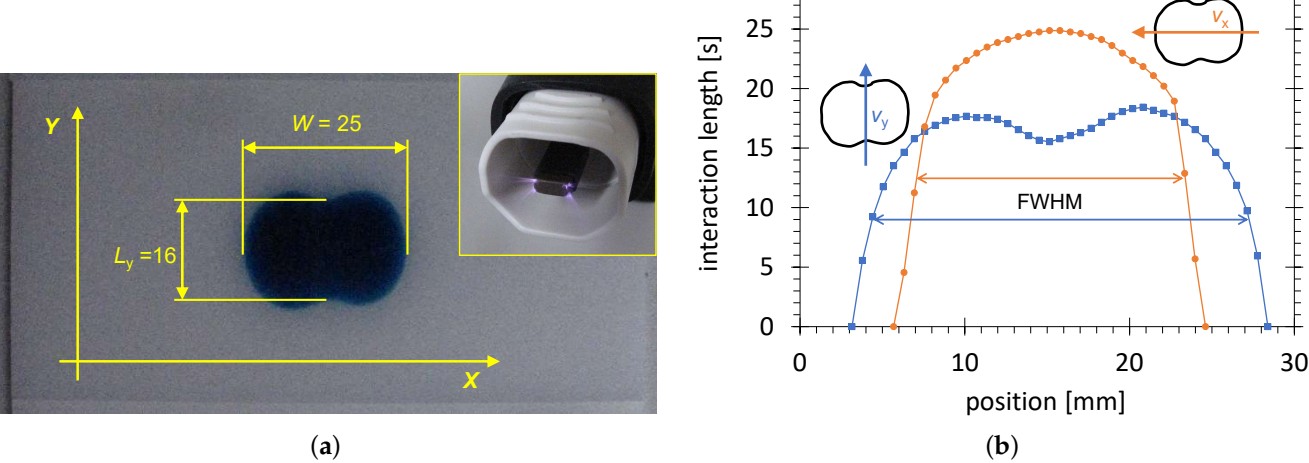

(**a**)                                                    (**b**)

**Figure 5.** The activated zone produced on a HDPE substrate by CeraPlas™ F module operated 10 s with the power of 8 W. (**a**) The image achieved statically by use of 58 mJ/m² test ink. (**b**) The interaction length of the activation zone with the single point of the substrate for movement of the substrate along the *x* and *y* axis of the ink image. The arrows show the movement direction with respect to the activation image. The point position is defined on the axis perpendicular to the movement direction. For both curves, the full widths at half maximum (FWHM) is depicted.

Assuming the same interaction time of 10 s by linear movement of the PCPG (the treatment time for the static treatment) the speeds of 1.6 and 2.25 mm/s for *y* and *x* direction respectively are achieved. The activation rate calculated by use of Equation (3) is in both cases the same: 1.6 mm/s × 22.5 mm = 2.25 mm/s × 16 mm = 36 mm$^2$/s.

This difference in the activation profile has practical implications for the design of PCPG arrays, allowing either faster activation on narrower substrates or slower activation on wider substrates. Using the results of the electric measurements the minimum distance between PCPGs assuring no interferences between these devices is $d_{\mathrm{min}} = 40$ mm, as depicted with red arrows in Figure 6a. To reach a homogeneous treatment when moving in *y* direction, two rows of PCPGs are sufficient as illustrated in Figure 6a. For movement along the *x* axis, at least three rows of PCPGs would be needed.

The size of the activation image depends strongly on the treatment time. Consequently, the width and length of the activation image varies as well. The curve with red triangles in Figure 6b illustrates the increase of the activation width $L_x$ as a function of treatment time. A similar curve can be plot for $L_y$ of the activation image, as defined in Figure 5a. Assuming the motion of the substrate in the *y* direction, the speed $v_y$ of this movement can be determined, by which the mean interaction time is equivalent to the static treatment time $t_{\mathrm{treat}}$:

$$v_{\mathrm{y}} = \frac{L_{\mathrm{y}}}{t_{\mathrm{treat}}}. \tag{4}$$

Taking into account the decrease of the activation length with decreasing treatment time, the dependence of the treatment speed is calculated and plotted in Figure 6b as the curve with blue circles. From this curve, it can be seen that the activation rate can be increased by an increase of the linear speed in the considered activation time range. It can also be shown, that if the PCPG movement speed $v_y$ would be higher than some limiting value the two rows of PCPG would be not enough to assure a homogeneous treatment. The treatment strips would not overlap anymore. The black arrows in Figure 6b show that the treatment width would be below 20 cm, if the treatment speed would exceed 6.3 mm/s.

The estimation in this example is valid for HDPE and surface energy of 58 mN/m. In many practical examples, less demanding materials are used and lower surface energies are required, resulting in a larger activation image for a shorter treatment time. Consequently, much higher treatment speed, in the range of tens of cm per second, can be achieved.

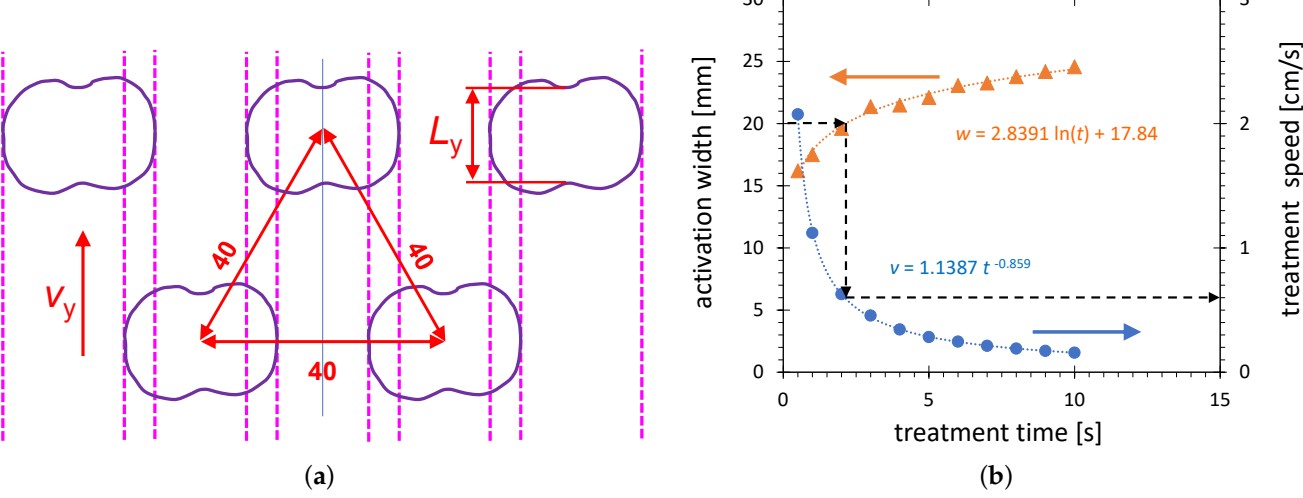

(**a**)        (**b**)

**Figure 6.** Determination of the CeraPlas™ F module array geometry. (**a**) The position of the CeraPlas™ F modules fulfilling the conditions of no interference and homogeneous treatment. (**b**) The width of the image of the activation area achieved on a HDPE substrate treated with piezobrush® PZ3 operated with power of 8 W and the equivalent treatment speed.

*3.5. Realization Examples*

3.5.1. Treatment of Substrates on a Belt Conveyor

In this example, the modules equipped with PCPG of CeraPlas™ F type are used to arrange an array for treatment of substrates moved on a belt conveyor with a width of 20 cm.

The substrates placed on a belt conveyor should be treated by PDD plasma. Using the PCPG configuration from Figure 6a the 10 PCPGs are sufficient to cover such treatment width. The maximum treatment speed of the substrates transferred by a belt conveyor is limited by the maximum PCPG power of 8 W. The only means of treatment rate scaling is to use a larger number of PCPG in series. In our case, four two-row arrays of 10 PCPGs each are applied (see Figure 7) allowing to increase the maximum speed of homogeneous treatment from 6.3 mm/s to about 25 mm/s.

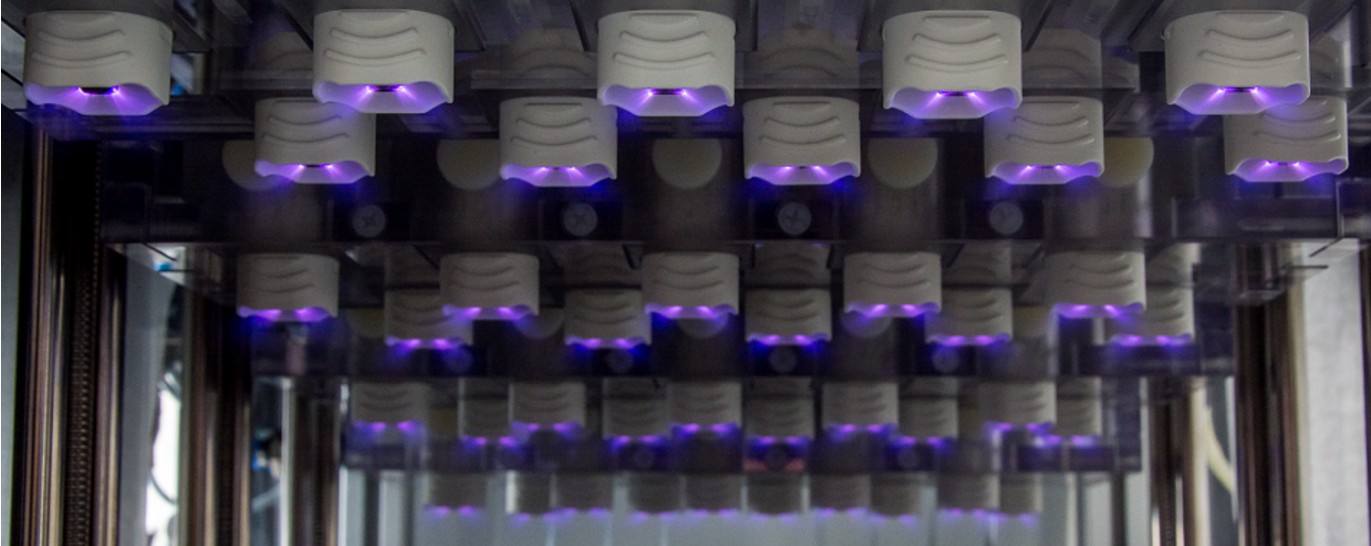

**Figure 7.** The plasma system consisting of 40 PCPGs organized into four units of 10 PCPG modules is shown under operation. The input power and CDA flow of each PCPG is 8.0 W and 7 SLM respectively.

The typical operating parameters of this system are the PCPG power of 480 W and the compressed dry air (CDA) consumption of 280 SLM. With these parameters, the ozone production rate is 3.2 g/h. The CeraPlas™ drives are supplied with 24 V DC from a common power supply. The entire system consists of input, treatment, and output chambers. Those are separated from each other and from the outside environment with silicon rubber curtains, which minimize the outflow of oxidizing species to the environment and keep their concentration in the treatment chamber high. The most important measure for avoiding the outflow of oxidizing species is the extraction system attached to the input and output chambers. The exhaust flow of the extraction system is adjusted to the gas intake of the PCPG modules. The PCPG modules are fed with CDA by means of the gas distribution system, with common gas manifold and mass flow controller. The belt conveyor transfers the substrates through the system. The adjustable vertical position of the PCPG module rows allows for the treatment of objects with very different sizes.

3.5.2. Precleaning of 6-Inch Silicon Wafers

The example presented in Section 3.5.1 refers to the treatment of electrically non-conducting or weak conducting substrates. If the substrate material is electrically-conductive, the PDD exists in a spark mode, with total energy focused on a small spot. It is disadvantageous if we aim for a large homogeneous area of treatment. The precleaning of highly-dopped silicon wafers described in this section is such a case with high treatment homogeneity requirement. The plasma focusing can be avoided by the application of a dielectric barrier discharge powered by the PDD. Different configurations of the PDD-

powered DBDs are described in [17]. The DBD configuration used in this example is known in the literature as floating electrode dielectric barrier discharge (FE-DBD) [29,30]. In such configuration, the substrate (in our case the silicon wafer) plays the role of the DBD passive electrode (Figure 8a). The DBD plasma is generated by the use of a coupling electrode which is biased from the PCPG (CeraPlas™ F) over a plasma bridge. The kHz oscillation of the coupling electrode is transferred capacitively to the air gap between the dielectric barrier and the surface of the silicon wafer. The DBD is sustained under the entire bottom surface of the coupling electrode, as shown schematically in Figure 8a).

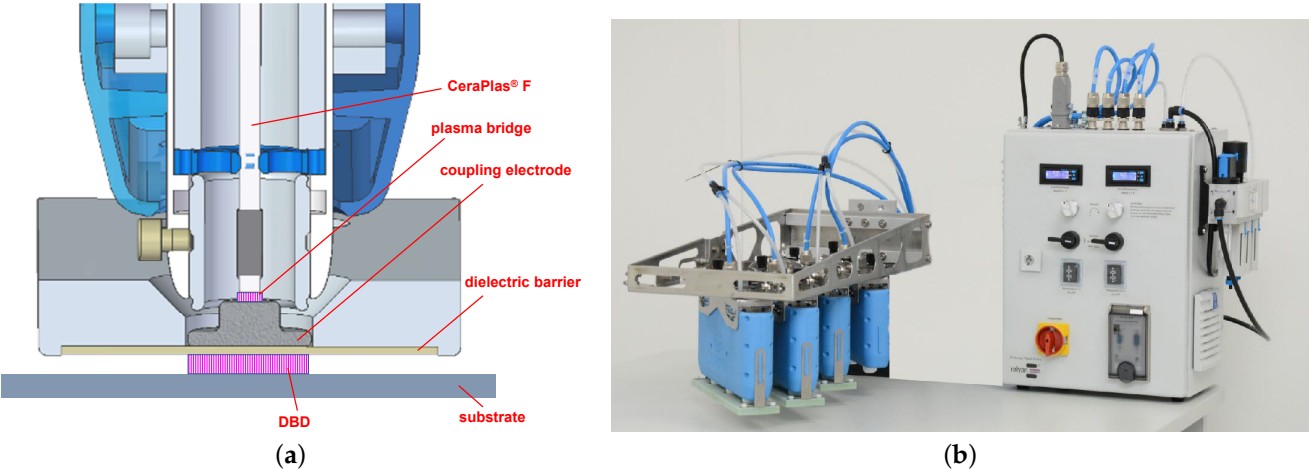

(**a**) (**b**)

**Figure 8.** The modular treatment unit with 12 CeraPlas™ F plasma sources organized in 4 sub-units and the power and gas control unit. (**a**) Plasma generation principle. (**b**) Picture of the system.

The size of the coupling electrode results from a compromise between two opposing tendencies. With increasing diameter of the coupling electrode, (i) the treatment width increases allowing higher activation rate but (ii) the efficiency of the DBD decreases resulting in the decrease of the activation rate. The reason for the second effect is the decrease of the oscillation voltage of the coupling electrode due to increasing the load capacity of the PCPG. Since the power of the PCPG is controlled by frequency, current and voltage, the increase of capacity resulting in increase of the blind current results in the decrease of voltage, causing less energetic micro-discharges in the DBD. Concluding, there exists an optimum diameter of the coupling electrode and it is, for the given shape and dielectric barrier, about 16 mm.

Figure 8b shows the realization of a system moving across a six-inch wafer assuring the homogeneous treatment. In Figure 9, schematically, the traces of 9 PCPGs produced by this system are drawn. This realization required the minimum PCPG tip-to-tip distance of 9 cm, which is assured when the red circles with the diameter of 9 cm are not overlapping. This tit-to-tip distance is larger than for example in Section 3.5.1 because the control driver of older type was applied, causing an audible noise for tip-to-tip distance up to 8 cm. The reason for such behavior was the swiping of a larger frequency range with a larger frequency step. In the currently used CeraPlas™ drive the multiparameter control is used. The input power of the piezoelectric transformer is controlled by changes of the input current, input voltage and the input signal frequency, resulting in moderate, non-periodic variations of the frequency.

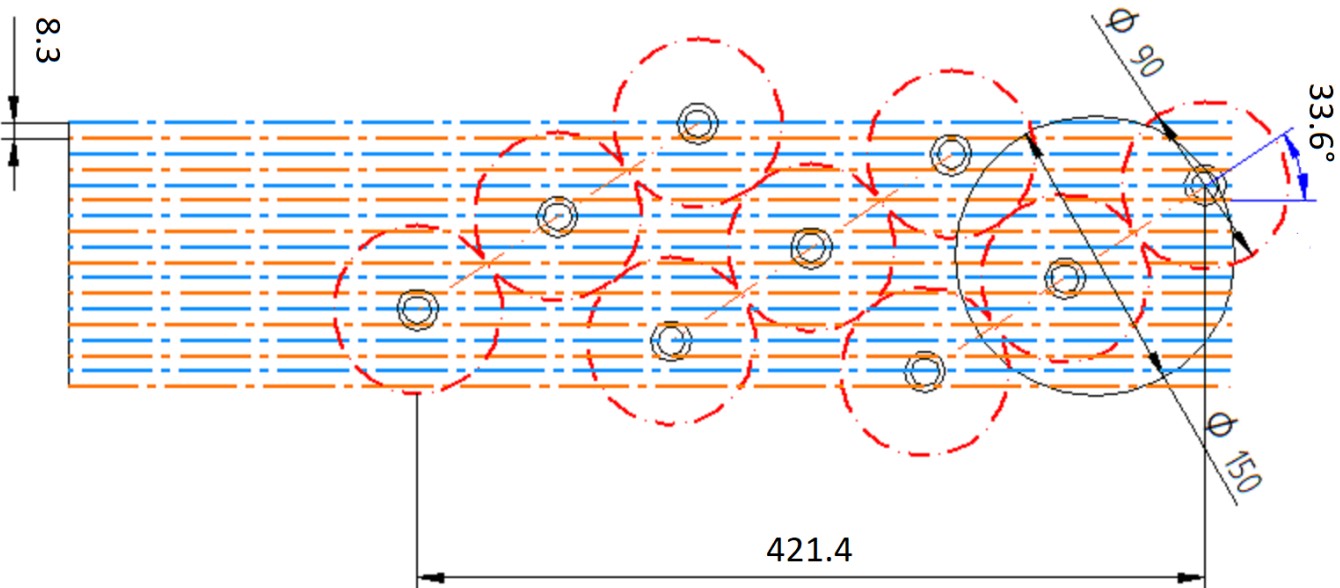

**Figure 9.** The traces of 9 PCPG-powered DBDs, each with width of 16.6 mm. The 6-inch wafer is treated in two sweeps with 8.3 mm offset: back and forth represented by blue and orange dot-dash-lines respectively. The black circles show the position of the coupling electrodes. The red dashed line circles visualize the minimum allowed tip-to-tip distance between the PCPGs.

## 4. Conclusions

The influence of the distance between an active and a passive PCPG on energy coupling between them is investigated. The minimum distance for safe operation is determined. It is 30 mm for the CeraPlas™ HF, 40 mm for the CeraPlas™ F powered by new type CeraPlas™ driver, and 90 mm for CeraPlas™ F powered by old type driver. The geometric criteria for homogeneous treatment with an array of PCPGs are investigated. Since the maximum activation width for CeraPlas™ F, defined as reaching the free surface energy of 58 mN/m on HDPE, is 25 mm, at least two rows of devices with overlapping activation patterns are needed to assure the treatment homogeneity, if the distance between the PCPGs is 40 mm. The activation width is decreasing with treatment speed. If some limiting speed value is exceeded, an additional row of PCPGs must be added to assure the homogeneous activation. The modularity concept and the calculation of the PCPG matrix layout is illustrated by two applications: the treatment of planar 8-inch substrates on a belt conveyor and the precleaning of 6-inch silicon wafers.

**Author Contributions:** Conceptualization, S.N., D.K., D.B., F.H., A.S. (Anatoly Shestakov), A.S. (Andrej Shapiro), and T.A.; Data curation, F.H., A.S. (Andrej Shapiro), T.A., and D.K.; Formal analysis, A.S. (Anatoly Shestakov) and D.K.; Funding acquisition, S.N. and S.L.; Investigation, A.S. (Anatoly Shestakov), D.K., A.S. (Andrej Shapiro), F.H., and D.B.; Methodology, D.K. and A.S. (Anatoly Shestakov); Project administration, S.L., A.S. (Andrej Shapiro), and F.H.; Resources, A.S. (Andrej Shapiro), F.H., D.B., and T.A.; Supervision, S.L.; Validation, S.N., D.K., and F.H.; Visualization, D.K., A.S. (Andrej Shapiro), and F.H.; Writing—original draft, D.K.; Writing—review & editing, D.K., S.L., F.H. All authors have read and agreed to the published version of the manuscript.

**Funding:** This research received no external funding.

**Institutional Review Board Statement:** Not applicable.

**Informed Consent Statement:** Not applicable.

**Data Availability Statement:** The data can be obtained on request from the first author.

**Acknowledgments:** The PCPGs used in this study: CeraPlas™ F and CeraPlas™ HF are provided by TDK Electronics GmbH. The authors thanks Jonas Wagner for electrical assembling of the multi-PCPG plasma devices.

**Conflicts of Interest:** The authors declare no conflict of interest.

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
