# Peer review of "Multi-Device Piezoelectric Direct Discharge for Large Area Plasma Treatment"

_plasma, doi:10.3390/plasma4020019_

Round 1
Reviewer 1 Report
This manuscript includes the following contents.
- Development of the piezoelectric cold plasma generators that produce multiple piezoelectric direct discharge using PZTs.
- For large-area plasma processing, the effective spacing between arrays by measuring the interference and coupling between two neighboring PCPGs.
- Brief description of three industrial applications using this device: (i) pretreatment of threads and fibers for coating, (ii) treatment of planar 8-inch substrates on a belt conveyor, and (iii) pre-cleaning of 6-inch silicon wafers.
In my opinion, the manuscript needs major revisions before publication as follows.
- First of all, the manuscript is not a journal article, but rather like a technical note on the design and performance of products related to the author's group company. The main reason this manuscript is likely to be considered a technical note is that the section 3.4 (Realization examples) occupies 5 out of 10 pages in the manuscript body. Readers are more interested in the contents of Interferences and coupling efficiency between PCPGs than device applications, so the reviewer recommend reducing the content of section 3.4.
- The references include conference proceedings and technical reports and there are several materials that cannot be verified as references. Replace conference proceedings and articles with journal articles as far as possible, in the references. Replace References with materials in English that are accessible to readers in the world.
- In Figures 3 and 4, the unit of time (x-axis) cannot be samples. And for better comparison, make the voltage range (y-axis) of the four graphs in Figures 3 and 4 the same.
- What is the relation between the probe voltage and the voltage that actually generates plasma? What is the output voltage that actually generates plasma?
- Move Figure 5 to a supplementary figure or remove it. I don't know which part of Figure 5(a) is enlarged to (b). It would be easier to understand to show the schematics instead of the picture.
Author Response
See the attached document

Reviewer 2 Report
I suggest the publication of this paper, after minor revisions. My comments are listed below.
1) Lines 16 – 28. An index is present. In my opinion an index is important for a book, but not for a research article. Moreover, is a repetition of the end of the introduction where (lines 54 – 57) the summary of the article is already presented. Therefore, delete the index (lines 16-28), and, after keywords, start with Introduction.
2) Line 36. The sentence: “Such array sources powered by 13 kHz, ….” is correct for reference 15, but reference 14 is relative to frequency range 1…100 kHz, and does not seem to be only related to 13 kHz. Therefore, I suggest to change the sentence in a form like “Such array sources powered with frequency in the kHz range, ….”
3) Figure caption of Figure 1. In my opinion, instead of “(b) capacitive probe voltage” is better “(b) capacitive probe measurement set-up”.
4) line 76. Indication “5000000” in my opinion is unreadable. I suggest to re-write the line in the compact form: “oscilloscope (5 M record length; 2.5 GS/s sample rate).”
5) Line 125. Change “on” with “a” in the sentence, i.e. “in hard PZT is on factor ….” with “in hard PZT is a factor …”
6) Figure 3 and Figure 4. On horizontal axis, use “time [μs]” and not “time [samples]”, and change the values in agreement. Scale values on horizontal axis and vertical axis: increase the font size for a better reading.
7) Line 178, at the end. “the” is present twice: delete one.
8) Line 225. In order to make the text clearer, I suggest to change “… how the width reduction” with “…how the reduction of the activation width”
9) Figure 7b has something wrong. Relations “w=87.4ln(t)+241” and “v=2.302 …” are not in agreement with left and right scales. For example. With t= 2 s, it results v = 1.4 that is not in agreement with the values on the vertical axis on the right. With t = 2 s, w = 301.6 that is not in agreement with the values on the vertical axis on the left. The figure must be corrected. As a further minor point, the two relations above indicated are written with a font size that should be increased for a better reading (if after correction relations are still present in the figure).
10) Figure 7b figure caption. It is mentioned the activation width and not the activation speed: add an indication to the activation speed.
11) References. Ref. 11 and ref. 19: doi indication is done twice. Ref. 27 is wrong: correct “…2020/10/201024…” with “…2020/11/201024…”. Ref. 16: update access.
Author Response
See the attached document

Round 2
Reviewer 1 Report
I have received answers to my comments, and the answers have been included in the manuscript.